# A microRNA Prognostic Signature in Patients with Diffuse Intrinsic Pontine Gliomas through Non-Invasive Liquid Biopsy

**DOI:** 10.3390/cancers14174307

**Published:** 2022-09-02

**Authors:** Maria F. Iannó, Veronica Biassoni, Elisabetta Schiavello, Andrea Carenzo, Luna Boschetti, Lorenza Gandola, Barbara Diletto, Edoardo Marchesi, Claudia Vegetti, Alessandra Molla, Christof M. Kramm, Dannis G. van Vuurden, Patrizia Gasparini, Francesca Gianno, Felice Giangaspero, Piergiorgio Modena, Brigitte Bison, Andrea Anichini, Sabina Vennarini, Emanuele Pignoli, Maura Massimino, Loris De Cecco

**Affiliations:** 1Fondazione IRCCS Istituto Nazionale dei Tumori, Via Amadeo 42, 20133 Milan, Italy; 2Pediatric Unit, Fondazione IRCCS Istituto Nazionale Dei Tumori, Via Venezian 1, 20133 Milan, Italy; 3Molecular Mechanisms Unit, Department of Research, Fondazione IRCCS Istituto Nazionale dei Tumori, Via Amadeo 42, 20133 Milan, Italy; 4Pediatric Radiotherapy Unit, Fondazione IRCCS Istituto Nazionale dei Tumori, Via Venezian 1, 20133 Milan, Italy; 5Human Tumors Immunobiology Unit, Department of Research, Fondazione IRCCS Istituto Nazionale dei Tumori, Via Venezian 1, 20133 Milan, Italy; 6Division of Pediatric Hematology and Oncology, Department of Child and Adolescent Health, University Medical Center Goettingen, Robert Koch Straße 40, 37075 Goettingen, Germany; 7Department of Neuro-Oncology, Princess Máxima Center for Pediatric Oncology, 3584 CS Utrecht, The Netherlands; 8Tumor Genomics Unit, Department of Research, Fondazione IRCCS Istituto Nazionale dei Tumori, Via Venezian 1, 20133 Milan, Italy; 9Department of Radiological, Oncological and Anatomo-Pathological Sciences, University Sapienza of Rome, 00185 Rome, Italy; 10IRCCS Neuromed, 86077 Pozzilli, Italy; 11Genetics Unit, Pathology Department, ASST Lariana General Hospital, 22100 Como, Italy; 12Department of Diagnostic and Interventional Neuroradiology, University Hospital Augsburg, 86156 Augsburg, Germany; 13Radiotherapy Department, Fondazione IRCCS Istituto Nazionale dei Tumori di Milano (INT), 20133 Milan, Italy

**Keywords:** neuro-oncology, circulating miRNAs, diffuse intrinsic pontine gliomas, prognosis

## Abstract

**Simple Summary:**

Diffuse intrinsic pontine glioma (DIPG) is a neuro-radiologically defined tumor of the brainstem, primarily affecting children, with most diagnoses occurring between 5 and 7 years of age. Surgical removal in DIPGs is not feasible. Subsequent tumor progression is almost universal and no biomarker for predicting the course of the disease has entered into clinical practice so far. Under these premises, it is essential to develop reliable biomarkers that are able to improve outcomes and stratify patients using non-invasive methods to determine tumor profiles. We designed a study assessing circulating miRNA expression by a high-throughput platform and divided patients into training and validation phases in order to disclose a potential signature with clinical impact. Our results for the first time have proved the usefulness of blood-circulating nucleic acids as powerful, easy-to-assay molecular markers of disease status in DIPG.

**Abstract:**

Diffuse midline gliomas (DMGs) originate in the thalamus, brainstem, cerebellum and spine. This entity includes tumors that infiltrate the pons, called diffuse intrinsic pontine gliomas (DIPGs), with a rapid onset and devastating neurological symptoms. Since surgical removal in DIPGs is not feasible, the purpose of this study was to profile circulating miRNA expression in DIPG patients in an effort to identify a non-invasive prognostic signature with clinical impact. Using a high-throughput platform, miRNA expression was profiled in serum samples collected at the time of MRI diagnosis and prior to radiation and/or systemic therapy from 47 patients enrolled in clinical studies, combining nimotuzumab and vinorelbine with concomitant radiation. With progression-free survival as the primary endpoint, a semi-supervised learning approach was used to identify a signature that was also tested taking overall survival as the clinical endpoint. A signature comprising 13 circulating miRNAs was identified in the training set (*n* = 23) as being able to stratify patients by risk of disease progression (log-rank *p* = 0.00014; HR = 7.99, 95% CI 2.38–26.87). When challenged in a separate validation set (*n* = 24), it confirmed its ability to predict progression (log-rank *p* = 0.00026; HR = 5.51, 95% CI 2.03–14.9). The value of our signature was also confirmed when overall survival was considered (log-rank *p* = 0.0021, HR = 4.12, 95% CI 1.57–10.8). We have identified and validated a prognostic marker based on the expression of 13 circulating miRNAs that can shed light on a patient’s risk of progression. This is the first demonstration of the usefulness of nucleic acids circulating in the blood as powerful, easy-to-assay molecular markers of disease status in DIPG. This study provides Class II evidence that a signature based on 13 circulating miRNAs is associated with the risk of disease progression.

## 1. Introduction

H3K27 altered diffuse midline glioma (DMG) is a rare group of malignancies included in the 2016 WHO Classification of Tumors of the Central Nervous System (CNS) and validated in the fifth one [1]. It refers to gliomas originating in the thalamus, brainstem, cerebellum, and spine with a dismal prognosis that has persisted despite the biomedical revolutions of the last century [1,2]. These tumors harbor somatic mutations in the H3F3A or HIST1H3B/C genes, resulting in lysine to methionine substitutions at amino acid residue 27 (K27M) in the histone H3 variants H3.3 or H3.1. The DMGs that infiltrate the pons are called diffuse intrinsic pontine gliomas (DIPGs), and they primarily affect early school-aged children. They are characterized by a rapid onset of symptoms in a previously healthy child [3]. DIPG is usually diagnosed based on the patient’s symptoms and magnetic resonance imaging (MRI). The tumor is typically hyperintense on T2-weighted imaging and on fluid-attenuated inversion images, while on T1-weighted imaging, the tumor is rather hypo- or isointense [4,5].

DIPGs originate intermixed with healthy tissue in the pons, a region of the brainstem containing many structures crucial to basic bodily functions; hence, it cannot be removed surgically [6,7]. Radiotherapy (RT) is effective for palliation in most cases, achieving transient improvements in neurological function and a progression-free survival (PFS) benefit, representing currently the mainstay of treatment [8]. In the last 30 years, DIPG patients have participated in more than 250 clinical trials worldwide, testing RT in combinations with a variety of chemotherapy drugs and small-molecule targeted inhibitors (alone or together with other drugs) [9,10,11,12]. A retrospective meta-analysis found that adjuvant systemic therapy was associated with a longer survival than radiation alone [13]. When tumor progression occurs, the median survival is reportedly 11 months, the overall survival proportion is 10% at 2 years, but less than 2% at 5 years after undergoing RT and any other adjuvant treatments [14,15].

Research suggests that neoplastic DIPG cells release a wide array of soluble molecules, some of which may enter the bloodstream [16]. Accordingly, these might serve as markers of response to therapy, or enable a new, functional classification of this tumor of potential relevance to its clinical course. In an effort to improve patient outcomes, it is essential to develop better prognostic tools and to better stratify cases by using alternative ways to ascertain tumor profiles.

Following an experimental workflow that included a discovery and a separate validation phase, we explored the potential of noninvasive blood biomarkers for improving the prognostic stratification of DIPG patients in terms of their risk of progression. We hypothesized that pursuing this approach could unveil a noninvasive biomarker of clinical utility in cases of DIPG, to better orient patients’ clinical management and ultimately, hopefully, improve their chances of survival.

## 2. Materials and Methods

### 2.1. Standard Protocol Approvals and Patient Consents

This study involved patients with DIPGs, who were treated between 2009 and 2017 at a referral center for pediatric solid tumors (Fondazione IRCCS Istituto Nazionale dei Tumori, Milan, Italy (INT)) and who were enrolled in either of two trials that combined nimotuzumab and vinorelbine with concomitant RT. This study thus had 47 cases: 23 in the pilot phase [17], which formed the “training set”, and 24 cases from the DIPG-INT 2015 (EudraCT: 2015-002185-23, 29 July 2015; ClinicalTrials.gov: NCT03620032, 2 November 2015), which served as the “validation set”. 

The methods were performed in accordance with relevant guidelines and regulations and approved by the local Ethical Committee of INT that approved the study design (INT 07/12 and INT 94/15); all parents, legal guardians or patients (if over 18 years old) signed to their informed consent to the use of their biological material and data for research purposes. 

### 2.2. Clinical Endpoints

The primary clinical endpoint of our investigation was progression-free survival (PFS). PFS times were calculated in months from the date of diagnosis to the date of any radiological or clinical evidence of progression, or death, due to the disease, and censored at the date of latest follow-up for patients still progression-free and alive. Disease progression was defined as established by the RAPNO working group [18]: (i) neurological deterioration confirmed by MRI (≥25% increase compared with smallest measurement at any timepoint from baseline in the 2D product of the perpendicular diameters using T2-weighted or FLAIR sequences); (ii) global deterioration in a participant’s physical condition not attributable to other causes, regardless of the radiological assessment. Clinical re-evaluation by the radiation oncologist and/or pediatric oncologist/neurologist was performed according to the standard practice. When pseudo-progression was suspected, a patient was retained in the study until disease progression was definitive, but the date of disease progression was backdated to the initial questionable progression timepoint if progression was ultimately confirmed on subsequent assessments. All images were also centrally reviewed both at diagnosis and during all the treatment phases by an expert neuroradiologist that was external to our protocols and not aware of clinical status. Tumor biopsies were not compulsory and performed only in the case of doubtful images.

Grade III, IV and V adverse events were registered using the CTCAE (Common Terminology Criteria for Adverse Effects), version 4.0.

As a secondary clinical endpoint, our ct-miRNA model was tested for its ability to identify differences in overall survival (OS). OS times were calculated in months from the date of diagnosis to the date of death due to the disease, and censored at the date of latest follow-up for patients that were still alive or who died of other causes.

### 2.3. Statistical Analysis

We first derived a survival model using the training data set. A biomarker complex score based on ct-miRNA expression and related to PFS was identified using a standardized, semi-supervised principal component method devised by Bair and Tibshirani [19]. 

Our proposed biomarker’s ability to predict survival risk was examined non-parametrically by using Kaplan–Meier curves, for which statistical significance between the at-risk patient groups (i.e., low vs. high risk) was assessed with the log-rank test. A univariate Cox’s proportional hazards regression was used to analyze the relationship between our ct-miRNA signature and PFS or OS. We also implemented a multivariate Cox’s regression in a follow-up analysis to determine whether our model provided predictions that were more accurate than, and independent from, the two covariates (i.e., patient age and hydrocephalus). Model checking and performance (goodness-of-fit) were assessed in terms of its (i) prediction error, based on Brier scores; (ii) calibration; (iii) discrimination; (iv) decision curve analysis. 

These analyses, sample processing, as well as other methods and materials above, are described in more detail in the Appendix A.

## 3. Results

### 3.1. Study Sample: Recruitment and Clinical Characteristics

Serum samples were collected at the baseline from 47 DIPG patients treated at INT in Milan from 2009 to 2017 and who were enrolled in a pilot study and in the control arm of the DIPG-INT 2015 (EudraCT 2015-002185-23). The samples were divided into training (*n* = 23) and validation (*n* = 24) data sets as explained earlier (Consort Diagram in Appendix A). 

The demographic and clinical characteristics of our two patient data sets are shown in Table 1. There were no significant differences found between the training and validation sets in terms of their age, gender, hydrocephalus (at diagnosis or during the course of the disease), or their pattern of cancer progression.

### 3.2. Development of a ct-miRNA Signature

All serum samples involved in our investigation of circulating miRNA profiles were checked for hemolysis levels. No hemolysis was observed in any of the samples considered, as confirmed by the low hemolysis scores (HS) for all 47 patients (Appendix A) from the spectrophotometric analysis. The training and validation sets had similar HS (Appendix A).

A high-throughput microRNA screening approach was used to identify miRNA profiles in serum samples at the baseline in the training data set. Data analysis yielded a matrix containing 293 detectable circulating miRNAs. To develop a prognostic model associated with PFS as the main clinical endpoint, a semi-supervised method of risk prediction was applied. This generated a signature containing 13 miRNAs, the first principal component of which retained 83.04% of the variation in their expression. Since we intended to ascertain whether circulating miRNA expression could predict the PFS, we considered the linear combination of our 13 miRNAs as a prognostic biomarker. 

We assessed skewness and kurtosis to examine the shape of the distribution of the miRNA index in order to distinguish multimodal distributions or outliers for possible exclusion. The data distribution of the miRNA index shows a skewness of 1.00 (*p* = 0.031) and a kurtosis of 4.34 (*p* = 0.07); the normality of data distribution was checked by the Shapiro–Wilk test (*p* = 0.0892). These properties were confirmed by the kernel density estimation of the joint distribution function of the biomarker, with the elapsed time being the time-to-event variable (Figure 1A). Since no relevant multimodal shapes were detected and the residuals were normally distributed (Appendix A), the model was developed to stratify patients using Leave-One-Out Cross-Validation (LOO-CV) applying the median signature value as the threshold, as detailed in Appendix A. The model segregated patients as being at high or low risk of progression, setting the dividing threshold at 0.007481 (Figure 1B). In the training set, 12 cases (54.2%) were classified as low risk, and 11 (45.8%) as high risk. As Figure 1C shows, the group predicted to be at high risk had a significantly shorter PFS than the group predicted to be at low risk (log-rank test, *p* = 0.00014; Fleming–Harrington test, *p* = 0.00328): the median PFS was 6 and 10.2 months for the high- and low-risk groups, respectively. Figure 1C shows the Kaplan–Meier analysis for the cross-validated risk groups. Our training set, however, included two long-term survivors, so we tested the impact of these two cases on our risk stratification: after excluding the long-term survivors, the performance of the model remained statistically significant (Appendix A).

### 3.3. Independent Validation of Our ct-miRNA Signature

To check our ct-miRNA model’s performance, 24 serum samples collected at the baseline from DIPG patients—enrolled in the first arm of the INT-DIPG2015 with assessable clinical data and an adequate follow-up—were used as a validation data set. Their serum samples were profiled for ct-miRNA expression in the same way as for the training data set. The expression of the 13 circulating miRNAs in the previously identified signature was also detectable in the validation set. We were thus able to derive a risk value for each sample in our validation set by applying our dividing threshold. The data distribution shows similar properties of the training set with a skewness of 0.341 (*p* = 0.418), a kurtosis of 4.27 (*p* = 0.082), and Shapiro–Wilk test equal to *p* = 0.318, supporting the normality of the distribution. The kernel density estimation of the biomarker, with time as the time-to-event variable, showed no relevant multimodal distributions (Figure 2A). To classify patients by their risk of relapse, we applied the cutoff obtained in our training phase (0.007481), which divided the cases into 12 patients at a high risk and 12 at a low risk of progression (Figure 2B). Kaplan–Meier curves confirmed the significantly different PFS for the two risk groups thus identified (log-rank test, *p* = 0.00026; Fleming–Harrington test, *p* = 0.0031 (Figure 2C), corresponding to a median of 7 and 10 months for the high- and low-risk groups, respectively. We then examined our ct-miRNA model’s ability to stratify patients in our validation set based on the OS; its Kaplan–Meier analysis showed a significantly different OS for the two risk groups (log-rank, *p* = 0.0021; Fleming–Harrington test, *p* = 0.00841 (Figure 2D)), corresponding to a median OS of 11.4 and 16.7 months for the high- and low-risk groups, respectively.

Univariate and multivariate regression models were used to assess the prognostic power of the ct-miRNA model compared with other covariates (age and hydrocephalus). Univariate analysis indicated that only the ct-miRNA model significantly predicted PFS and OS. When all the covariates were analyzed simultaneously in a multivariate model, the ct-miRNA model maintained its significant predictive ability for PFS, whereas now the presence of hydrocephalus and the ct-miRNA model were both independent, significant prognostic factors for OS (Table 2).

### 3.4. Performance of Our ct-miRNA Signature

To test the utility of our proposed signature in clinical practice, and to see whether its merit was not simply an artifact of our data sets’ small sample sizes, the model’s performance was quantified using the validation set. For this, a significant relationship between the patients’ risk class and PFS was confirmed in the Kaplan–Meier analysis. We derived estimates for continuous and/or binary response models, quantifying how close predictions came to actual (observed) outcomes. In particular, we considered: (i) measures of overall performance, using the Brier score; (ii) discrimination, using sensitivity and specificity metrics; (iii) calibration, using plots of predicted vs. observed outcomes; (iv) clinical benefit, based on a decision curve analysis. For this analysis, our estimates took into account the model’s performance over time, or, alternatively, we used the median follow-up time point for the validation set (=8.5 months).

To corroborate the validity of our model, an overall estimate was explored first, and the prediction error of our model in fitting the survival information was examined with the Brier score (Figure 3A). These results showed that the expected Brier score was lower than the reference scenario’s score when it took the risk identified by the ct-miRNA model into account. The LOO-CV estimate of the Brier score for the ct-miRNA signature at the median time of 8.5 months was 0.126, while that for the reference scenario was 0.270 when the patients were not stratified. This supports the ct-miRNA model’s worthiness, since when the Brier score approaches zero the closer predicted values fit the actual (observed) ones at each follow-up time point. Finally, as a summary measure of the Brier scores, the cumulative prediction error (IBS) over an interval ranging from 0 to 16 months was 0.055 for the ct-miRNA signature, which was almost half the 0.106 value achieved for the reference scenario. 

The calibration of a model refers to the agreement between the predicted outcome of interest and the observed outcome. Here, we found that the calibration line tracks closely the 1:1 correspondence (diagonal), suggesting a reasonable agreement between the probability estimated by the ct-miRNA model and the actual PFS (Figure 3B). Hence, this indicated the predictions are adequate, and so is the model’s performance.

Discrimination analysis of the derived risk model generated the sensitivity and specificity of our model, which was assessed by generating a ROC curve. The AUC at the landmark follow-up time of 8.5 months had a value of 0.96 (95% CI: 0.888–1) (Appendix A) with the sensitivity, specificity, positive predictive value (PPV) and negative predictive value (NPV) equal to 88.9, 81.8, 80, and 90%, respectively. Nevertheless, the AUC only indicates the discriminatory ability of our model at a given time point; so, a time-dependent ROC curve was calculated for our ct-miRNA model to take into account the censoring pattern of patients over the whole period (Figure 3C). To assess the practicability of the ct-miRNA model, a DCA was fitted to our data to reveal the potential net benefits of the model for clinical decision-making. Viewed graphically, the DCA demonstrated the ct-miRNA model has a positive net benefit for predicted probability thresholds in the range between 1 and 54%, corresponding to the intercept between the two reference conditions (i.e., all true negative, and all true positive rates) (Figure 3D). For very low threshold probabilities (<10%), however, in which patients receive treatment despite a relatively low risk of progression, the net benefit is marginally higher if patients are stratified by the ct-miRNA model, but this is insufficiently valuable to improve their care. However, for threshold probabilities ranging from 10 to 54%, a decision based on the ct-miRNA model’s results is a superior option. For threshold probabilities of >54%, a threshold at which unnecessary investigations and treatments may be a concern, the option offered by the ct-miRNA model holds a significant value.

Likewise, discrimination and calibration examinations were made to establish the prognostic performance of the model’s fit but with OS as the endpoint, which further confirmed its merit (Appendix A).

## 4. Discussion

DIPGs are rapidly growing tumors associated with dismal survival. Biopsy is difficult and may be dangerous to obtain due to the anatomical location of these tumors (i.e., the brainstem). In addition, concerns about the information provided by biopsy specimens have arisen, since the representativeness of tumor heterogeneity and the inability to cover the course of the disease, providing only a snapshot at the time of resection, impairs their utility [20]. An alternative to tissue biopsies is liquid biopsies, requiring minimally invasive procedures to analyze nucleic acids or proteins in blood and cerebrospinal fluid (CSF) for the tumor-specific genetic signatures. For instance, the driver mutation H3K27M can be detected in circulating DNA (ctDNA) from peripheral blood, enabling reliable monitoring during and after treatment with a decreased content of ctDNA if the tumor had receded [21]. 

MicroRNAs are non-coding small RNA molecules that can be secreted into the circulation and exist in remarkably stable forms, representing a remarkable opportunity in liquid biopsy translational research [22]. In the present study, we have pursued an ongoing exploratory project that focused on miRNAs in blood with two goals in mind: improve our understanding of DIPGs and avoid the need for invasive and complicated routine biopsies in the young patients affected. 

We designed and conducted a study on a total of 47 homogeneously-treated patients, applying a signature development framework, and dividing the cases into a training data set (*n* = 23 patients) and a separate validation data set of a nearly equal size (*n* = 24). Our initial findings from the former were confirmed by the latter, hence we had identified a ct-miRNA signature that could be used in prospective projects and trials to stratify patients by their high or low risk of progression, or as a surrogate of tumor resistance or sensitivity to RT. Our sample size (i.e., number of patients) was limited when compared with the large number of features assessed (i.e., 2006 miRNAs initially checked for), so we considered the issue of overfitting in our data analysis. This happens when a model fits its behavior in a training set to the extent that it negatively affects its performance with new data, prompting an unrealistic overestimation. To address this issue, we estimated the predictive performance of our ct-miRNA model in the validation set. This was done using various methods and traditional measures of survival outcomes, including the Brier score (to indicate a model’s overall performance), the area under the ROC curve, and goodness-of-fit statistics for calibration. Decision curve analysis (DCA) was also reported to assess if a predictive model could be useful for clinical decisions. DCA combines a clinical intuition regarding the usefulness of a diagnostic testing with an assessment of whether the test is really worth performing. In fact, the decision curve applied to our ct-miRNA model suggested a relative net benefit when patients were segregated by the model for predicting the probability of progression at an 8.5-months follow-up.

Considering the dismal prognosis for patients with DIPGs, identifying those with a temporarily better or worse prognosis within the same trial might seem pointless. As pediatric oncologists, however, it is also our ethical responsibility to ensure our patients (especially those with a poor prognosis) do not receive useless or even toxic/painful treatments. The innovative clinical importance of our ct-miRNA signature lies in the fact that it enables us to predict which patients will respond poorly to RT, and thus avoid further courses of this treatment, or re-irradiation at relapse, which has become almost a standard for patients with recurrent DIPG [23]. 

Additional tests would be required to determine the true nature of each of the miRNAs identified here, by exploring their behavior in vivo. We identified a signature with 13 circulating miRNAs, but little is known about their biological functions in neurological diseases (Table 3) [24,25,26]. The overexpression of miR-4714-3p, miR-551b and miR-4505 is related to a better prognosis in our patients. miR-4505 is reportedly involved in the nervous system, midbrain development, and nerve growth factor receptor signaling pathways, and its overexpression has been associated with the onset of generalized anxiety disorder [27]. MiR-551b is overexpressed in gliomas, while miR-4714-3p has been reported to be dysregulated in patients with multiple sclerosis [24,28]. The overexpression of the remaining 10 miRNAs of our signature (i.e., miR-6090, miR-6089, miR-3960, miR-936, miR-1207-5p, miR-202-3p, miR-3676-5p, miR-4634, miR-4539, and miR-4299) is related to a worse prognosis. All 10 miRNAs are involved in various biological processes, such as cell proliferation, gliomagenesis, pathological brain conditions, and radioresistance. Wang and colleagues investigated the role of miR-936 in glioma tissue specimens, providing evidence that miR-936 is correlated with tumor grade and a worse survival [25]. MiR-4299 is reported to be expressed in glioma cells and it influences the tumor microenvironment [26]. MiR-3676-5p has been investigated in pituitary adenomas, where it plays a role in regulating genes involved in tumor invasiveness [29]. Some of the miRNAs we found—particularly miR-6090, miR-4505, miR-6089, miR-3960, miR-1207-5p, and miR-4634—were reportedly overexpressed in patients with intracerebral hemorrhage, so they may be related to a pathological condition of brain tissues [30]. Recent studies have revealed that differences in miRNA expression can influence radiosensitivity in various tumors, including glioblastoma, and miR-4539 is linked to radioresistance in atypical meningioma patients [31,32].

Since the blood–brain barrier impacts the release of putative biomarkers into the systemic circulation, it has been hypothesized that the cerebrospinal fluid (CSF) can serve as a source of biological material reflecting the brain physiological and pathological conditions better when compared to peripheral blood [33]. Thus, the analysis of the 13 circulating miRNAs in CSF is warranted to enable more precise prognostic approaches. Although the signature was validated in an independent validation set, a multicenter prospective study should be designed to confirm its prognostic value; for this purpose, activities are required to transpose the signature into a useful clinical grade assay following the guidelines defined by the Institute of Medicine [34]. and REMARK [35].

## 5. Conclusions

Our results ought to be combined with those obtained in tissues or fluids (serum or CSF) by other research groups. Liquid biopsy techniques are expected to provide a relevant molecular landscape of the disease. Ideally, the integration of clinical status, imaging characteristics, and liquid biopsy-based molecular characterization will offer a novel monitoring approach able to provide a comprehensive clinical and molecular snapshot of the tumor in space and time. Given the paucity of available tissue, and the generally low incidence of the disease, greater collaborative efforts to improve its prognosis have been one of the few positive notes in recent decades in the clinical world of DIPG. Not-for-profit foundations have sponsored its research, leading to the creation of DIPG registries pooling clinical, radiological, pathological, and molecular data. Thanks to these efforts, DIPG research continues to grow, and with it sustaining new hope for the future.

## Figures and Tables

**Figure 1 cancers-14-04307-f001:**
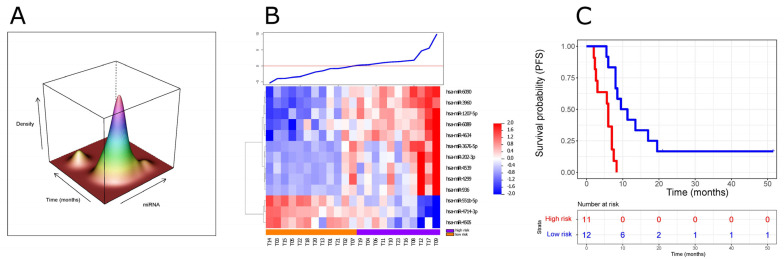
Prognostic signature based on ct-miRNAs from the training data set. (**A**) Joint density estimation for the ct-miRNA signature and time-to-event variable in the training set (*x*-axis = miRNA model; *y*-axis = time-to-event; *z*-axis = density kernel estimate). (**B**) Heatmap of the expression levels related to the 13 miRNAs entering the signature. On the horizontal axis are the respective miRNAs; on the vertical axis are the patient samples (*n* = 23). The samples are ranked based on the signature score, having the dividing threshold at 0.007481 defining those of a low or high risk, and the line plot above the heatmap summarizes the score value per sample. (**C**) Kaplan–Meier survival curves for patients predicted to be at high (blue, *n* = 11) or low (red, *n* = 12) risk of cancer progression. High-risk patients had a shorter PFS (progression-free survival) than those at low risk (log-rank test, *p* = 0.00014; hazard ratio (HR) = 7.99, 95% confidence interval (CI) 2.38–26.87). The permutation test (based on 100 permutations) had a *p*-value of 0.03, indicating a low probability of overfitting for the above-mentioned log-rank analysis. The Schoenfeld individual test was assessed to test Cox regression assumption and to discard any violation considering the fast dip to zero trend for the high-risk cases in contrast to the low-risk cases. Since the Schoenfeld individual test reaches *p* = 0.911, the test is not statistically significant and, therefore, we can assume the proportional hazards.

**Figure 2 cancers-14-04307-f002:**
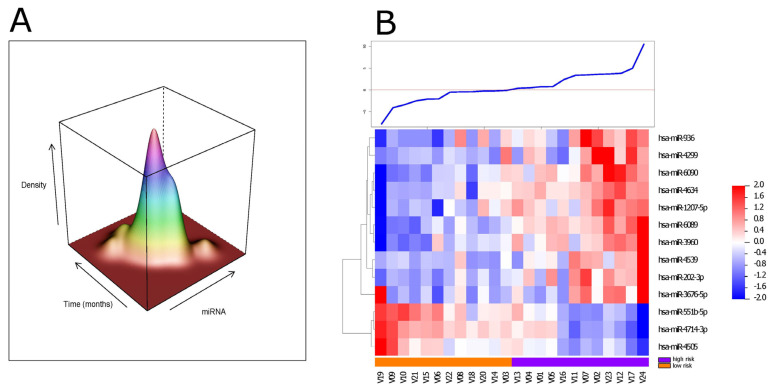
Validation of the ct-miRNA signature. (**A**) Joint density estimation for the signature and time-to-event variable in the validation data set (*x*-axis = miRNA model; *y*-axis = time-to-event; *z*-axis = density kernel estimate). (**B**) Heatmap of the expression levels of the 13 miRNAs comprising the signature. On the horizontal axis are the respective miRNAs; on the vertical axis are the patient samples (*n* = 24). Samples are ranked based on the signature score; even if the rank order differs somewhat between the two heatmaps (training vs. validation), the division between the miRNAs remains clear; the line plot above the heatmap summarizes the score value per sample. (**C**) Kaplan–Meier survival curves for patients predicted to be at a high (blue, *n* = 12) or low (red, *n* = 12) risk of progression, taking PFS (progression-free survival) as the endpoint. High-risk patients had a significantly shorter PFS than those at a low risk (log-rank test, *p* = 0.00026; hazard ratio (HR) = 5.51, 95% confidence interval (CI) 2.03–14.9). (**D**) Kaplan–Meier survival curves taking OS (overall survival) as the clinical endpoint. When risk stratification by the ct-miRNA model was tested for OS, it was significantly shorter for high-risk than low-risk patients (log-rank, *p* = 0.0021, hazard ratio (HR) = 4.12, 95% confidence interval (CI) 1.57–10.81). High-risk in red, low-risk in blue.

**Figure 3 cancers-14-04307-f003:**
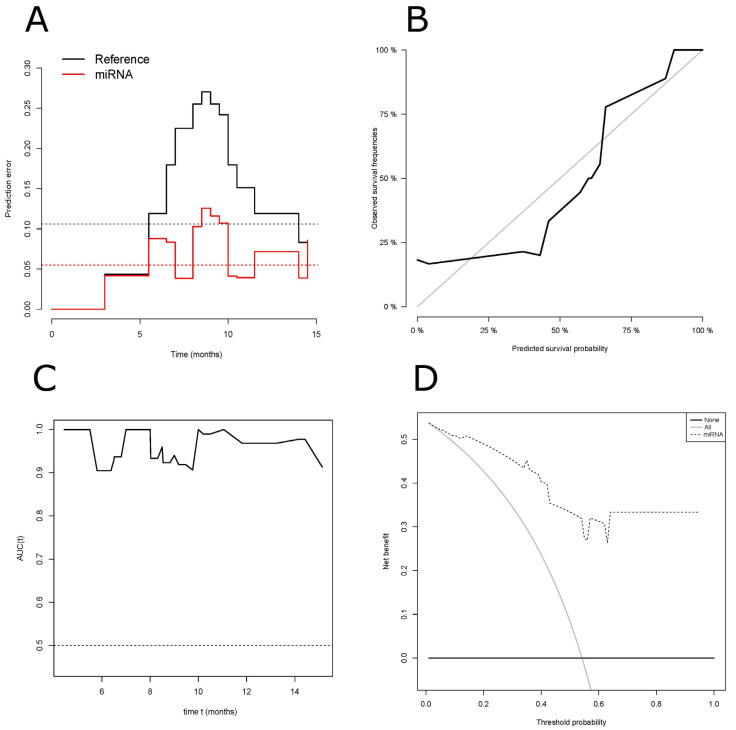
Assessment of the model’s performance. (**A**) Prediction error curves for Brier scores based on the miRNA signature stratification, and also on estimates for all patients without any stratification applied (reference scenario curve). For a single patient, the Brier score at the time t is defined as the squared difference between the observed survival status and the predicted outcome probability. Red dotted line = IBS for the ct-miRNA model; black dotted line = IBS for the reference scenario. (**B**) Calibration plot for PFS (progression-free survival) at the landmark follow-up time point of 8.5 months. The plot shows the predictions obtained by the model on the *x*-axis and the observed outcomes on the *y*-axis. (**C**) Area under the ROC curves (AUC) based on our ct-miRNA model’s fitting of the PFS, obtained from a time-dependent ROC analysis. (**D**) DCA (decision curve analysis) for the model’s efficacy at predicting progression in DIPG, and to assess the clinical utility of the proposed miRNA model. DCA shows, graphically, the clinical usefulness of the ct-miRNA model based on a continuum of potential thresholds for patients’ risk of progression (*x*-axis) and the net benefit of using the ct-miRNA model to stratify patients (*y*-axis). The horizontal black line indicates the all-true negative rate (corresponding to the risk assumption that no patients had disease progression at 8.5 months) and the diagonal gray line indicates the all-true positive rate (corresponding to the risk assumption that all patients had disease progression at 8.5 months). The dotted line indicates an improvement in the prediction achieved by the ct-miRNA model.

**Table 1 cancers-14-04307-t001:** DIPG tumor cohorts.

		Training Set (*n* = 23)	Validation Set (*n* = 24)	*p*-Value
**Age (median, range)**		6.68 (2–17 y)	7.07 (2–21 y)	0.512 ^Ŧ^
**Sex**	Male	10	12	0.654 ^ŦŦ^
	Female	13	12	
**Hydrocephalus**	Yes	4	8	0.21 ^ŦŦ^
	No	19	16	
**Pattern of progression**	Local	14	20	0.145 ^ŦŦ^
	Disseminated	7	4	
	No progression	2	0	

^Ŧ^ *p*-value for a two-sample *t*-test; ^ŦŦ^ *p*-values for χ2 test for the contingency table.

**Table 2 cancers-14-04307-t002:** Results of Cox’s proportional hazard regression analysis.

	Univariate Analysis (PFS)		Multivariate Analysis (PFS)	
PFS	HR (95% CI)	*p*-value	HR (95% CI)	*p*-value
Hydrocephalus (presence vs. absence)	0.807 (0.33–1.971)	0.638	1.481 (0.517–4.246)	0.465
Age	0.9926 (0.92–1.07)	0.849	1.009 (0.935–1.09)	0.825
ct-miRNA (high vs. low risk)	5.506 (2.034–14.9)	**0.000786**	6.525 (2.129–20.0)	**0.00103**
	**Univariate analysis (OS)**		**Multivariate analysis (OS)**	
OS	HR (95% CI)	*p*-value	HR (95% CI)	*p*-value
Hydrocephalus (presence vs. absence)	1.936 (0.787–4.759)	0.15	2.8751 (1.111–7.44)	**0.0295**
Age	0.998 (0.925–1.076)	0.961	0.994 (0.922–1.072)	0.8846
ct-miRNA (high vs. low risk)	4.119 (1.57–10.81)	**0.0042**	5.351 (1.939–14.771)	**0.0012**

*p*-Values < 0.05 are in bold; HR, hazard ratio; 95% CI, 95% confidence interval.

**Table 3 cancers-14-04307-t003:** Literature information.

Gene Id	Weights (*w_i_*)	Circulating miRNA in Liquid Biopsy	Involment in Neurological Diseases	Suggested/Documented Functional Role in Neurological Disease	References	Reported in Other Tumors	Suggested/Documented Functional Role in Tumor Other Than Brain	References
**hsa-miR-4714-3p**	−0.889482	Reported	blood from patients with multiple sclerosis	not investigated	Keller, 2014 [28]	Head-Neck squamous cell carcinoma	not investigated	Huang Y, 2020 [36]
**hsa-miR-6090**	0.401593	Reported	cerebrospinal fluid from patients with intracerebral haemorrhage	pathological condition of brain	Stylli, 2017 [30]	downregulatd in Multiple Myeloma patients	not investigated	Zhang, 2019 [37]
**hsa-miR-4505**	−0.402474	Reported	nervous system	nervous system development, nerve growth factor receptor signaling	Chen, 2016 [27]	Myeloma Patients	downregulation is associated with progression of disease	Zhang, 2019 [37]
			cerebrospinal fluid from patients with intracerebral haemorrhage	pathological condition of brain	Stylli, 2017 [30]	metastatic-intramucosal carcinoma patients	not investigated	Kim S, 2020 [38]
			patients with generalized anxiety disorder	not investigated	Wu, 2018 [24]	upregulated in colon cancer pantients	not investigated	Wang, 2017 [39]
**hsa-miR-551b-5p**	−0.850107	Reported	glioblastoma tissue	not investigated	Wu, 2018 [24]	downregulated in Gastric Cancer patients	regulation of ubiquitin-dependent protein catabolic process, cell division, and mRNA stability	Jiang X, 2019 [40]
**hsa-miR-6089**	0.54622	Reported	cerebrospinal fluid from patients with intracerebral haemorrhage	pathological condition of brain	Stylli, 2017 [30]	Ovarian Cancer	promotes cancer cell proliferation, migration, invasion and metastasis	Liu L, 2020 [41]
**hsa-miR-3960**	0.431525	Reported	cerebrospinal fluid from patients with intracerebral haemorrhage	pathological condition of brain	Stylli, 2017 [30]	downregulated in Bladder Cancer patients	not investigated	Usuba, 2018 [42]
**hsa-miR-936**	0.170501	Not reported	glioma tissue	downregulation is associated to worse overall survival	Wang, 2017 [25]	nasopharyngeal carcinoma	not investigated	Wang 2020 [43]
**hsa-miR-1207-5p**	0.466562	Reported	cerebrospinal fluid from patients with intracerebral haemorrhage	pathological condition of brain	Stylli, 2017 [30]	gastric cancer tissues	downregulation promote proliferation, invasion and induces cell cycle arrest in gastric cancer cells in vitro and in vivo	Chen L, 2014 [44]
**hsa-miR-202-3p**	0.345363	Reported	glioma tissue	involvement in cell proliferation, migration and proliferation	Yang, 2017 [45]	differentially expressed in cervial cancer	not investigated	Yi, 2016 [46]
**hsa-miR-3676-5p**	0.151234	Reported	pituitary adenoma	regulation of tumor suppressor genes involved in invasion	Wu S, 2017 [29]	lung cancer	not investigated	Qin, 2017 [47]
**hsa-miR-4634**	0.46722	Reported	cerebrospinal fluid from patients with intracerebral haemorrhage	pathological condition of brain	Stylli, 2017 [30]	non-small cell lung cancer cells	overexpression is associated with better prognosis of NSCLC patients.	Liu S, 2020 [48]
**hsa-miR-4539**	0.066802	Reported	atypical meningioma	downregulation is associated to radioresistance	Zhang, 2020 [32]	gastric cancer patients	not investigated	Zhang C, 2018 [49]
**hsa-miR-4299**	0.069597	Reported	pediatric glioma stem cells exosomes	influence of tumor microenvironment/normal neural stem cells	Tuzesi, 2017 [26]	non-small cell lung cancer cells	overexpression inhibits the proliferation, migration and invasion in vitro	Yang, 2018 [50]

## Data Availability

The datasets generated and analyzed in this study are available in the Gene Expression Omnibus (GEO) repository (http://www.ncbi.nlm.nih.gov/geo/ (1 September 2022)) under GSE160249.

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
