# Peer review of "A microRNA Prognostic Signature in Patients with Diffuse Intrinsic Pontine Gliomas through Non-Invasive Liquid Biopsy"

_cancers, 2022, doi:10.3390/cancers14174307_

Round 1

Reviewer 1 Report

To editors and authors

Different prognosis in diffuse intrinsic pontine gliomas through non invasive liquid biopsies

This is a very interesting manuscript that should be considered for publication in CANCERS after some revisions below.

1) Please recheck and revise cautiously citation and references as MDPI format.

2) Add IRB date and number

3) In introduction part, need to mention about diagnostic imaging. Please check and refer updated document (PMID: 33137983)

4) Figures are quite small. Please try to revise and enhance.

5) Limitation part is unclear. Please revise along with further direction study.

Sincerely

Author Response

This is a very interesting manuscript that should be considered for publication in CANCERS after some revisions below.

We thank the reviewer for his/her positive comments and suggestions that help us to improve the quality and hopefully the general readability of our manuscript.

1) Please recheck and revise cautiously citation and references as MDPI format.

R1: the references were reformatted following MDP1 format

2) Add IRB date and number

R2: the IRB IDs were added (i.e. INT 07/12 and INT 94/15 for training and validation sets, respectively)

3) In introduction part, need to mention about diagnostic imaging. Please check and refer updated document (PMID: 33137983)

R3: the introduction section has been implemented mentioning the role of diagnostic imaging in DIPG and the suggested reference has been added

4) Figures are quite small. Please try to revise and enhance.

R4: High resolution figures (600 DPI) figures have been uploaded separately

5) Limitation part is unclear. Please revise along with further direction study.

R5: we thank the reviewer for addressing the issue. A paragraph has been added in discussion pointing out the limitations of our study that should require additional validation in a multicenter study and the biological functional analysis of the miRNAs included in the signature to foster its clinical application.

Reviewer 2 Report

The group led by Massimino present a very interesting and potentially important contribution to the field of non-invasive liquid biopsy for patients with the incurable childhood brain tumour, diffuse intrinsic pontine glioma (DIPG), a subset of diffuse midline gliomas (DMG) as per WHO CNS5 (2021). The authors have identified a set of 13 microRNAs (miR) in a training cohort of 23 patients and subsequently in a validation cohort of 24 patients treated similarly. This miR signature can clearly distinguish patients with very poor (high risk) and poor (low risk) prognosis for both PFS (primary outcome) and OS (secondary outcome). Prospectively, this 13 miR signature may provide useful for clinicians to discuss with parents of affected children prior to initiating therapy.

A. Major Concerns

1. Results:

a. It is unclear from the data presented what is the actual sensitivity and specificity of the 13 miR signature that was validated in the second patient cohort of 24 patients.

2. Discussion:

a. The authors should briefly discuss what is known about the 13 miRs that make up the prognostic signature. This information with references is already provided by the authors in the legend to Table S1. This information belongs in the main manuscript, so this request is readily remedied.

b. The authors should also discuss the current status of liquid biopsy from serum and CSF for the purposes of non-invasive diagnosis of specific Histone 3.3/3.1 mutations and/or disease surveillance and/or monitoring response to therapy. In this manner the identified 13-miR prognostic signature can be placed in the best context for use in the clinic.

B. Specific Concerns

1. Title: The title should be rewritten to include "microRNA signature". One suggestion is: "A microRNA prognostic signature obtained from patients with diffuse intrinsic pontine gliomas through non-invasive liquid biopsy".

2. Abstract: Need to state explicitly that the serum sample was obtained at the time of MRI diagnosis and prior to radiation and/or systemic therapy.

3. Introduction:

a. Line 79: As per WHO CNS5 (2021), the correct term is H3K27 altered diffuse midline glioma.

b. Line 82: Rewrite to mention the specific H3.3 K27M and H3.1 K27M gene mutations.

4. Materials and Methods:

a. Line 205: Define the term "LOO-CV" at first use and refer the reader to the Supplementary Methods (page 5) for further details.

5. Results

a. Can the authors provide any further information regarding the 2 long-term survivors from the training cohort?

b. Was CSF tested for the miR signature? for presence of the H3K27M mutation?

6. Discussion

a. What is known about the predicted gene/miR targets of the identified miRs in the signature?

b. Is it possible that radiation and/or systemic therapy could alter the microRNA signature, thereby affecting its use to follow disease progression as a means of surveillance?

c. Please comment on whether there might be clinical utility to re-check the miR signature after RT, the end of systemic therapy (if offered) or as a means of disease surveillance? Are there any plans to do this in future clinical trials?

C. Minor Concerns

a. Line 379: Replace "few" with "little".

b. Line 383: Add in parentheses after fluids "serum or CSF".

Author Response

The group led by Massimino present a very interesting and potentially important contribution to the field of non-invasive liquid biopsy for patients with the incurable childhood brain tumour, diffuse intrinsic pontine glioma (DIPG), a subset of diffuse midline gliomas (DMG) as per WHO CNS5 (2021). The authors have identified a set of 13 microRNAs (miR) in a training cohort of 23 patients and subsequently in a validation cohort of 24 patients treated similarly. This miR signature can clearly distinguish patients with very poor (high risk) and poor (low risk) prognosis for both PFS (primary outcome) and OS (secondary outcome). Prospectively, this 13 miR signature may provide useful for clinicians to discuss with parents of affected children prior to initiating therapy.

R: We thank the Reviewer for recognizing the interest of the topic of our study.

  1. Major Concerns
  2. Results:
  3. It is unclear from the data presented what is the actual sensitivity and specificity of the 13 miR signature that was validated in the second patient cohort of 24 patients.

R1: To determine the value of the 13-miRNA signature established in training set including the threshold of 0.007481, we analyzed an independent set from a cohort of 24 DIPG patients. We considered 8.5 months and 13.7 months as landmark timepoints to assess sensitivity and specificity, PPV, and NPV for PFS and OS, respectively, corresponding to the median progression and survival times. The landmark timepoints were consistent with the results of the pilot phase 2 study that combined nimotuzumab with concomitant radiation and vinorelbine (Massimino, M. et al. J Neurooncol 118, 305–312) reporting 8.5 and 15 months as median PFS and OS, respectively. Regarding the primary endpoint (PFS), the sensitivity, specificity, PPV, NPV, and AUC were 88.9%, 81.8%, 80%, 90%, and 0.96, respectively. For OS (secondary endpoint), the sensitivity, specificity, PPV, NPV, and AUC were 71.4%, 80%, 75%, 75%, and 0.778, respectively. The manuscript has been implemented accordingly.

  1. Discussion:
  2. The authors should briefly discuss what is known about the 13 miRs that make up the prognostic signature. This information with references is already provided by the authors in the legend to Table S1. This information belongs in the main manuscript, so this request is readily remedied.

R2a: The details provided in TableS1 has been moved to discussion as suggested.

  1. The authors should also discuss the current status of liquid biopsy from serum and CSF for the purposes of non-invasive diagnosis of specific Histone 3.3/3.1 mutations and/or disease surveillance and/or monitoring response to therapy. In this manner the identified 13-miR prognostic signature can be placed in the best context for use in the clinic.

R2b: we thank the reviewer for this comment and a section in discussion has been added and a prominent work on ctDNA demonstrating the feasibility of clinical utility of liquid biopsises in DIPG has been referenced (doi: 10.1158/1078-0432.CCR-18-1345).

  1. Specific Concerns
  2. Title: The title should be rewritten to include "microRNA signature". One suggestion is: "A microRNA prognostic signature obtained from patients with diffuse intrinsic pontine gliomas through non-invasive liquid biopsy".

RB1: We thank the reviewer for his/her suggestion and the title has been rewritten.

  1. Abstract: Need to state explicitly that the serum sample was obtained at the time of MRI diagnosis and prior to radiation and/or systemic therapy.

RB2: the abstract was been modified accordingly

  1. Introduction:
  2. Line 79: As per WHO CNS5 (2021), the correct term is H3K27 altered diffuse midline glioma.
  3. Line 82: Rewrite to mention the specific H3.3 K27M and H3.1 K27M gene mutations.

R3a and R3b: the manuscript has been implemented as suggested

  1. Materials and Methods:
  2. Line 205: Define the term "LOO-CV" at first use and refer the reader to the Supplementary Methods (page 5) for further details.

R4a: as suggested, at first use in the main text, “Leave-One-Out Cross-Validation” was added to explain the acronym and it was referenced to the methods available in supplementary methods.

  1. Results
  2. Can the authors provide any further information regarding the 2 long-term survivors from the training cohort?

R5a. We do not appreciate any significant difference that could clearly explain long-term survival compared to the rest of the cohort; this is also due to the low number of long-term cases in our case series. However, we tried to correlate the clinical features of the 2 long-term survivors with literature data. Thanks to an international effort, a huge study from Hoffman LM et al (10.1200/JCO.2017.75.9308) evaluated the clinical characteristics of 1,000 long-term centrally-reviewed DIPG cases. This study highlighted that about 10% of cases survivors are long term survivors (LTS). This is consistent with our data where LTS represent about 8% of cases. Hoffman LM et al evaluated the characteristics of LTS and age <3 years at presentation was more frequently associated with long-term survival. The median age at diagnosis of our training set is 7.8 yrs. The long term survivors have an age of 2.1 and 5.1 yrs,  respectively, below the median age and one was <3 years old. In addition, both patients presented with hydrocephalus. As reported in Massimino et al (doi.org/10.1007/s11060-014-1428-z) the onset of hydrocephalus is associated with a favorable prognosis in patients treated with nimotuzumab and concomitant radiation and vinorelbine. Although these evidences are inconclusive, they are in line with the literature.

  1. Was CSF tested for the miR signature? for presence of the H3K27M mutation?

R5b: CSF represents the ideal material for biomarker research due to its proximity to the tumors and an increasing number of studies proved the feasibility to detect nucleic acids and proteins in CSF of children. However, in the clinical practice as in our cohort, heterogeneity should be expected with the absence or presence of hydrocephalus and the need for CSF access or diversion during the course of treatment done with different methods (lumbar puncture, endoscopic third ventriculostomy, Ommaya reservoire  shunt). In addition, the collection and processing of the material also require the setting-up of standard operating procedures (SOPs) to avoid pre-analytical issues. Our study included only a handful of matched blood/CSF specimens where it was feasible to investigate the miRNAs of our signatures. Although we were able to detect the expression of our miRNAs in CSF with a consistent trend with the matched blood specimens, due to the limited sample size, the different methods in CSF access, and the absence of established SOPs, the data are not suitable to be reported in the present paper.

  1. Discussion
  2. What is known about the predicted gene/miR targets of the identified miRs in the signature?

R6a: The details provided in TableS1 has been moved to discussion as suggested (see also R2a)

  1. Is it possible that radiation and/or systemic therapy could alter the microRNA signature, thereby affecting its use to follow disease progression as a means of surveillance?

R6b. To the best of our knowledge, no information is available about the influence to ct-miRNAs  induced by radiation/systemic treatment in DIPG. We can infer some hypotheses based on the experience coming from other tumors. Growing evidence suggests that circulating nucleic acid concentration correlates with tumor burden, cancer stage, cellular turnover, and response to therapy (doi: 10.1177/1758835918794630). Literature data proved that the expression of miRNAs is altered following ionizing radiation at the peripheral blood level and serum miRNAs may be used to predict the biological impact of radiation doses playing a rule as clinically feasible biodosimeters. Researchers also identified several time-specific and dose-specific miRNAs (doi: 10.3389/fcell.2022.861451; doi: 10.3390/jpm10030072.).

This offers the foreground of investigating circulating miRNAs as biomarkers for optimized treatment decisions.  In this way, in order to improve radiotherapy, several aspects should be balanced and evaluated exploiting potential circulating miRNA signatures: (1) use of single and fractionated radiation protocols; (2) use of a gradient of doses; (3) extension of the assessment of temporal dynamics of miRNA expression.

In addition, various radiation and chemotherapy protocols have been attempted in DIPG patients and, at present, only conventional radiotherapy has yielded improvements in survival, so far. A study compared proton therapy versus conventional photon radiotherapy in terms of the outcomes but failed to demonstrate superior survival outcomes of proton therapy even  if the treatment was well tolerated by the majority of patients with no severe adverse events, including radiation necrosis (DOI: 10.1007/s00381-019-04420-9). In theory, proton therapy reduces the areas receiving unnecessary irradiation and providing a consistent advantage. In the future,  new radiation modalities will likely become the treatment of choice.  In this frame, circulating miRNA and our signature could be a useful tool to monitor the patients, avoiding late sequelae.

  1. Please comment on whether there might be clinical utility to re-check the miR signature after RT, the end of systemic therapy (if offered) or as a means of disease surveillance? Are there any plans to do this in future clinical trials?

R6c. Liquid biopsies have gained a clinical interest for monitoring DIPG disease course. At initial diagnosis, DIPG patients undergo magnetic resonance imaging (MRI), followed by invasive brain biopsy to determine diagnosis based on WHO classification paradigms. Despite the importance of MRI in long-term treatment monitoring, it can be difficult to distinguish between radiation treatment effects such as pseudoprogression, radiation necrosis, and recurrent/progressive disease based on imaging alone in the majority of patients who receive radiation therapy. Monitoring based on tissue biopsies is risky and not feasible. Nevertheless, making a clear distinction between these entities is crucial for patient treatment and has a big impact on survival. Liquid biopsy techniques are expected to provide a relevant molecular landscape of the disease. As a matter of fact, ideally, the integration of clinical status, imaging characteristics, and liquid biopsy based molecular characterization will offer a novel monitoring approach able to provide a comprehensive clinical and molecular snapshot of the tumor in space and time. Thus, the assessment of the tumor evolution and the identification of true progressions from radiation effects have the potential to meet the current challenges in DIPG. This warrants further translational research with the design of clinical studies including the exploitation of the potential of liquid biopsies. Discussion has been modified accordingly.

  1. Minor Concerns
  2. Line 379: Replace "few" with "little".
  3. Line 383: Add in parentheses after fluids "serum or CSF".

R6Ca and R6Cb: the text was been modified accordingly.

Round 2

Reviewer 2 Report

The authors have responded to all of this reviewer's concerns. the revised manuscript is significantly improved.

However, the revisions would have benefited from English editing. 

Also, Supplementary Table 1 should now be Table 3 and included with the revised manuscript (it remains with the Supplementary materials).

Minor revisions (all in the Discussion):

1. Line 356: Replace "is a" with "are". Replace "to" with "with".

2. Line 357: Add "may be" prior to "dangerous". The safety of brainstem tumour biopsy is well established and there are several references in the literature that attest to this.

3. Line 359: Delete "been".

4. Line 363: Replace "driven" with "driver".

5. Line 365: Delete "also a".

6. Line 404: Change "Supplementary Table 1" to "Table 3" and add this table into the main manuscript.

7. Line 405: Replace "are" with "is".

8. Line 418: Delete "the".

9. Line 431: Rewrite as "reflect brain physiology and related pathology".

Author Response

The authors have responded to all of this reviewer's concerns. the revised manuscript is significantly improved.

R1: We thank the reviewer and we are deeply grateful for his/her positive comments and for his/her thorough revision and suggestions that help us to improve the quality and hopefully the general readability of our manuscript.

However, the revisions would have benefited from English editing.

Also, Supplementary Table 1 should now be Table 3 and included with the revised manuscript (it remains with the Supplementary materials).

R2: The table has been moved from the supplementary material and now is Table3.

Minor revisions (all in the Discussion):

  1. Line 356: Replace "is a" with "are". Replace "to" with "with".
  2. Line 357: Add "may be" prior to "dangerous". The safety of brainstem tumour biopsy is well established and there are several references in the literature that attest to this.
  3. Line 359: Delete "been".
  4. Line 363: Replace "driven" with "driver".
  5. Line 365: Delete "also a".
  6. Line 404: Change "Supplementary Table 1" to "Table 3" and add this table into the main manuscript.
  7. Line 405: Replace "are" with "is".
  8. Line 418: Delete "the".
  9. Line 431: Rewrite as "reflect brain physiology and related pathology".

R3:  we thank the reviewer for the editing; the text has been revised as suggested.